# The Effect of Autotrophic Cultivation of *Platymonas subcordiformis* in Waters from the Natural Aquatic Reservoir on Hydrogen Yield

Magda Dudek [1], Marcin Dębowski [1,*], Anna Nowicka [1], Joanna Kazimierowicz [2] and Marcin Zieliński [1]

[1] Department of Environmental Engineering, Faculty of Geoengineering, University of Warmia and Mazury in Olsztyn, 10-720 Olsztyn, Poland; magda.dudek@uwm.edu.pl (M.D.); anna.grala@uwm.edu.pl (A.N.); marcin.zielinski@uwm.edu.pl (M.Z.)

[2] Department of Water Supply and Sewage Systems, Faculty of Civil Engineering and Environmental Sciences, Bialystok University of Technology, 15-351 Bialystok, Poland; j.kazimierowicz@pb.edu.pl

[*] Correspondence: marcin.debowski@uwm.edu.pl

**Abstract:** Biological processes run by microalgae are prospective but still little known methods of hydrogen production. A prerequisite for their increased advancement is the development of economically viable and efficient technologies. The study presented in this manuscript focused on determining the efficiency of biohydrogen production by *P. subcordiformis* using a culture medium prepared based on natural waters. The rate of *P. subcordiformis* biomass growth reached $317.6 \pm 42.3$ $mg_{ODM}/dm^3 \cdot d$ and ensured a biomass concentration of $3493 \pm 465$ $mg_{ODM}/dm^3$. The percentage concentration of hydrogen in the biogas reached $63.2 \pm 1.4\%$, and its production rate ranged from $0.53 \pm 0.05$ $cm^3/h$ to $0.70 \pm 0.01$ $cm^3/h$.

**Keywords:** *Platymonas subcoriformis*; microalgae; biohydrogen; photobioreactor; biomass; culture medium; natural waters

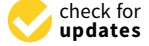



## 1. Introduction

The negative impact of the economy on the natural environment determines the need to use low-emission production technologies, including the implementation of clean and effective solutions for energy production [1,2]. Hydrogen represents one of the energy carriers meeting the criteria of an environmentally friendly fuel [3]. Today, it is used in a narrow and marginal range, mainly in the refining industry, space technologies and fuel cells [4,5]. The lack of rational, technologically and economically viable methods for its production and storage are the main hurdles for its large-scale deployment [6]. Conventional hydrogen production technologies include mainly thermochemical methods, like combustion, gasification, thermochemical liquefaction and pyrolysis, as well as methods based on water electrolysis [7]. However, these solutions entail high investment costs, are energy-consuming and cause environmental pollution [8,9]. It is estimated that currently nearly 95% of the hydrogen used derives from the conversion of fossil fuels [10].

The increasingly prospective hydrogen production technologies include biomass-harnessing solutions and methods based on biological processes conducted by microorganisms [11]. These include primarily the fermentation of organic substrates carried out by specialized groups of bacteria or biochemical processes taking place in cells of selected microalgal species [12,13]. Due to the very high photosynthetic efficiency, fast biomass growth rate, resistance to various types of pollutants, susceptibility to genetic modifications and the possibility of developing land that cannot be used for other purposes, it is the use of microalgae that becomes the most promising avenue of biohydrogen production [14,15].

One of the conditions necessary to achieve high hydrogen productivity by microalgae is the use of efficient biomass production technologies [16]. The most perspective solutions

for microalgae cultivation and proliferation include installations based on the use of sea waters, which are not curbed on a global scale neither by their limited amount nor by costs of their acquisition [17,18]. The use of waters from natural aquatic reservoirs has been proved viable in the cultivation of microalgae species with high adaptive and competitive abilities, resistant to diseases and parasites and not very sensitive to harsh and varying environmental conditions. These include microalgae of the genus *Chlorella* sp., *Scenedesmus* sp., and cyanobacteria [19,20]. In the case of taxa, the cultivation of which is focused on the production of specific products, preference is given to cultures in fully monitored closed photobioreactors, stable conditions and in appropriate culture media, usually composed of distilled water, nutrients, microelements, and vitamins [21,22].

The species used to produce hydrogen include mainly unicellular algae with specific metabolic and enzymatic traits allowing for its production [23], as for example, taxa belonging to green algae and cyanobacteria, mainly *Chlamydomonas reinhardtii*, and algae of the genus *Chlorella* sp. [24]. *P. subcordiformis* is also considered a very prospective species in the context of hydrogen production in the process of direct biophotolysis [25].

The study presented in this manuscript focused on determining the efficiency of biohydrogen production by *P. subcordiformis* species microalgae using a culture medium prepared based on waters from the coastal zone of the Baltic Sea.

## 2. Materials and Methods

### 2.1. Experimental Design

Research works were divided into two stages (ST). Experiments conducted in STAGE 1 (ST1) aimed to assess the efficiency of *P. subcordiformis* microalgae biomass production This stage included two series, differing in the culture medium applied. Deionized water was used in SERIES 1 (SE1), whereas water from the Gdańsk Bay was used in SERIES 2 (SE2). STAGE 2 (ST2) aimed to analyze the impact of the culture conditions applied on the hydrogen yield. It was analogously divided into two series (SE1 and SE2). In addition, each of these series was divided into two experimental variants differing in the initial concentration of algal cells in the bioreactor, that is, 3.0 $g_{ODM}/dm^3$ in VARIANT 1 (V1) and 5.0 $g_{ODM}/dm^3$ in VARIANT 2 (V2). Table 1 and Figure 1 present the scheme of research works.

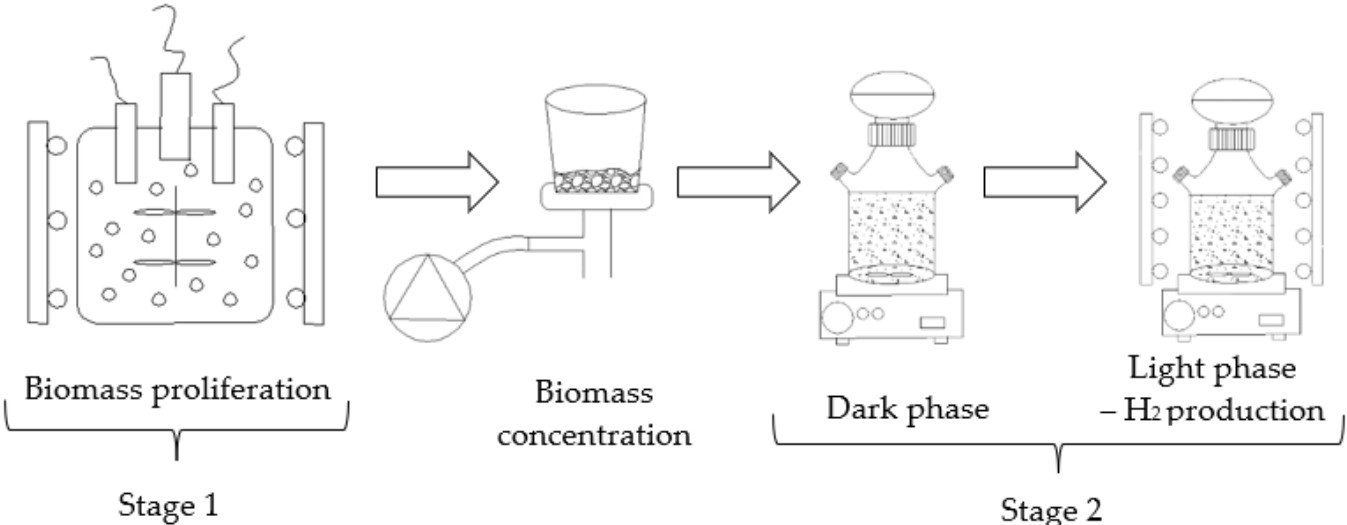

**Figure 1.** Scheme of experimental works.

**Table 1.** Experimental design.

| ST1—cultivation of *P. subcordiformis* | SE1—culture medium based on deionized water and chemical reagents. | ST2—H$_2$ production by *P. subcordiformis* | SE1 | V1—3.0 g$_{ODM}$/dm$^3$ |
|---|---|---|---|---|
| | | | | V2—5.0 g$_{ODM}$/dm$^3$ |
| | SE2—culture medium based on water from the Gdańsk Bay supplemented with chemical reagents. | | SE2 | V1—3.0 g$_{ODM}$/dm$^3$ |
| | | | | V2—5.0 g$_{ODM}$/dm$^3$ |

### 2.2. Materials

The *Platymonas subcordiformis* species microalgae derived from own collection grown based on the inoculum obtained from the UTEX algae culture collection UTEX (University of Texas at Austin, Austin, TX, USA). The initial concentration of microalgal biomass expressed as the concentration of organic dry matter reached 200 mg$_{ODM}$/dm$^3$ in ST1, whereas in ST2, it depended on the variant (Table 1).

In ST1SE1, the composition of the culture medium was as follows: 1.30 mg/dm$^3$ FeCl$_3$, 0.36 mg/dm$^3$ MnCl$_2$, 33.60 mg/dm$^3$ H$_3$BO$_3$, 45.00 mg/dm$^3$ EDTA, 20.00 mg/dm$^3$ NaH$_2$PO$_4$, 100.00 mg/dm$^3$ NaNO$_3$, 0.21 mg/dm$^3$ ZnCl$_2$, 0.20 mg/dm$^3$ CoCl$_2$, 0.09 mg/dm$^3$ (NH$_4$)$_4$Mo$_7$O$_{24}$, 0.20 mg/dm$^3$ CuSO$_4$, 0.10 μg/dm$^3$ VB12, and 1.00 μg/dm$^3$ VB1. Salinity fell within the range of 30–33 ppt, whereas pH within the range of 8.00–8.20. In ST1SE2, the composition of the culture medium was as follows: 59.97 ± 0.25 mg COC/dm$^3$, 0.04 ± 0.01 mg NH$_4$–N/dm$^3$, 21.85 ± 1.05 mg N$_{tot}$/dm$^3$, 2.95 ± 0.79 mg PO$_4^{3-}$/dm$^3$, 5.05 ± 0.31 mg P$_{tot}$/dm$^3$, 538 ± 2.52 mg SO$_4^{2-}$/dm$^3$, 7411 ± 59.5 mg Cl$^-$/dm$^3$, 2380 ± 156 mg Cl$^-$/dm$^3$, 0.107 ± 0.01 mg Fe$^{2+}$/dm$^3$, 0.087 ± 0.01 mg Fe$^{3+}$/dm$^3$, whereas pH was 8.04 ± 0.15 and salinity reached 30 ± 0.3 ppt. Water from the Gdańsk Bay was sampled from May until October. It was filtered through filters for qualitative analyses (medium size, Ø 12.5, Eurochem), and then sterilized in a Tuttnauer 2840 EL—D autoclave at a temperature of 121 °C for 15 min.

In ST2, deionized water served as the culture medium for hydrogen production in SE1. In SE2, hydrogen was produced using water from the Gdańsk Bay supplemented with the following culture medium: 27.23 mg/dm$^3$ NaCl, 5.079 mg/dm$^3$ MgCl$_2$, 1.123 mg/dm$^3$ CaCl$_2$, 0.667 mg/dm$^3$ KCl, 0.196 mg/dm$^3$ NaHCO$_3$, 0.098 mg/dm$^3$ H$_3$BO$_3$, 0.098 mg/dm$^3$ KBr, 0.024 mg/dm$^3$ SrCl$_2$, 0.003 mg/dm$^3$ NaF, and 0.002 mg/dm$^3$ CuCl$_2$, with pH ranging from 7.90 to 8.00.

### 2.3. Experimental Station

In ST1, the microalgal biomass was grown in BioFlo 115 New Brunswick bioreactors having an active volume of 2.0 dm$^3$, at a temperature of 25 ± 1 °C, white light with the intensity of 5 klux (14 h light/10 h dark cycle), aeration using a Mistral 200 pump with the capacity of 200 dm$^3$/h, and continuous stirring at 150 rpm. In all experimental series, *P. subcordiformis* algae were cultured for 11 days. The biomass from the BioFlo 115 reactor was concentrated deploying a kit for vacuum filtration consisting of a filter mounted in the MBS 1 filtration kit (Whatman) that enables separating algal biomass from the culture medium using a Mobile 20 vacuum pump.

In ST2, the concentrated biomass of *P. subcordiformis* was fed to respirometers having an active volume of 0.5 dm$^3$ (Wissenschaftlich-Technische Werkstätten (WTW), Weilheim in Oberbayern, Deutschland). Oxygen was removed by purging the respirometers with pure nitrogen for 3 min. The dark phase was ensured by tightly covering the bioreactors with aluminum foil. The retention time of *P. subcordiformis* biomass in the dark phase was 30 h. Afterwards, the reactors were illuminated with white light having the intensity of 5 klux, for 5 days. The contents of the reactors were stirred at 100 rpm using VMS—C4 Advanced magnetic agitators. The experiment was conducted at the temperature of 25 ± 1.0 °C.

*2.4. Analytical and Statistical Methods*

Samples were collected every 48 h. Contents of dry matter, organic dry matter (ODM), and mineral dry matter in the biomass were determined with the gravimetric method. Chlorophyll was determined spectrophotometrically after extraction with 90% acetone. Total nitrogen (EN ISO 11905-1), ammonia nitrogen (ISO 7150-1), total phosphorus (ISO 6878_2004), orthophosphates (ISO 6878_2004), sulphates (ISO 10304-1), chlorine compounds (ISO 9297:1994), iron compounds (DIN 38406-E1), and COD (ISO 6060-1989) were determined using Hach Lange cuvette tests and an UV/VIS DR 5000 spectrophotometer. The same methodology was applied to monitor the levels of essential nutrients in the culture, that is, $N_{tot.}$ and $P_{tot}$. The taxonomic analysis was performed using an MF 346 microscope with an Optech 3MP camera. The culture medium composition was controlled using a UV/VIS DR 5000 spectrophotometers (Hach Lange), salinity of the culture medium—using Marine Control Digital (Aqua Medic), and light intensity—using an HI 97500 luxometer (HANNA). The volume of the produced biogas was computed based on pressure changes inside the measuring chamber. The respirometric analysis also allowed the determination of the biogas production rate (r). Biogas quality was analyzed with the GC Agillent 7890 A gas chromatograph, and the analysis included determinations of the percentage contents of the following biogas fractions: carbon dioxide ($CO_2$), oxygen ($O_2$), and hydrogen($H_2$).

The research was carried out in four replications. The statistical analysis of experimental results was carried using a STATISTICA 13.1 PL package. One-way analysis of variance (ANOVA) was performed to determine the significance of differences between groups. HSD Tukey test was deployed to find significant differences between the analyzed variables. In the tests, results were considered significant at $p = 0.05$.

## 3. Results

In STSE1, the final biomass concentration reached $3203 \pm 35$ $mg_{ODM}/dm^3$, whereas chlorophyll a content in the biomass was at $3686 \pm 320$ $\mu g/dm^3$ (Table 2). The rate (r) of microalgal biomass growth reached $291.2 \pm 3.2$ $mg_{ODM}/dm^3 \cdot d$. In addition, the P and N compounds were almost completely depleted. At the end of the cultivation, their concentrations were at $0.61 \pm 0.37$ mg $N_{tot}/dm^3$ and $0.04 \pm 0.01$ mg $P_{tot}/dm^3$. In turn, the mean consumption of these biogenes per biomass growth unit was at $7.0 \pm 0.4$ mg $N_{tot}/g_{ODM}$ and $1.7 \pm 0.1$ mg $P_{tot}/g_{ODM}$ (Table 2). In ST1SE2, the rate of *P. subcordiformis* biomass growth reached $317.6 \pm 42.3$ $mg_{ODM}/dm^3 \cdot d$, which allowed reaching biomass concentration of $3493 \pm 465$ $mg_{ODM}/dm^3$. The concentration of chlorophyll a reached $3845 \pm 696$ $\mu g/dm^3$ and was produced at $r = 49.6 \pm 63.3$ $\mu g/dm^3 \cdot d$ (Table 2). The compounds of nitrogen and phosphorus were effectively removed from the culture medium as their final concentrations in the technological system reached $0.39 \pm 0.13$ mg $N_{tot}/dm^3$ and $0.02 \pm 0.01$ mg $P_{tot}/dm^3$.

**Table 2.** Results obtained in STAGE 1 of the study.

| | Indicator | | | | | | |
|---|---|---|---|---|---|---|---|
| **Series** | **Final Biomass Concentration [mg $_{ODM}$/dm³]** | **Biomass Growth Rater [mg$_{ODM}$/dm³·d]** | **Final Concentration of Chlorophyll a [µg/dm³]** | **Effectiveness of $N_{tot.}$ Removal [%]** | **Effectiveness of $P_{tot.}$ Removal [%]** | **$N_{tot.}$ Consumption for Biomass Growth [mg $N_{tot}$/g$_{ODM}$]** | **$P_{tot.}$ Consumption for Biomass Growth [mg $P_{tot}$/g$_{ODM}$]** |
| SE1 | 3203 ± 35 | 291.2 ± 3.2 | 3686 ± 320 | 98.1 ± 0.6 | 97.6 ± 1.1 | 7.0 ± 0.4 | 1.7 ± 0.1 |
| SE2 | 3493 ± 465 | 317.6 ± 42.3 | 3845 ± 696 | 98.4 ± 0.9 | 99.1 ± 0.4 | 6.7 ± 0.6 | 1.5 ± 0.1 |

In ST2SE1, the total volume of biogas produced in V1 reached $108.57 \pm 9.14$ $cm^3$ at $r = 0.91 \pm 0.08$ $cm^3/h$, whereas in V2—the *P. subcordiformis* biomass produced $141.58 \pm 6.31$ $cm^3$ of biogas with the rate of $r = 1.81 \pm 0.05$ $cm^3/h$ (Figures 2 and 3). In turn, hydrogen production reached $42.79 \pm 3.60$ $cm^3$ in V1 and $57.44 \pm 2.56$ $cm^3$ in V2 (Figures 2 and 3). The percentage content of hydrogen in biogas approximated 42% in both variants (Table 3). Regardless of the experimental variant, the biogas production per *P. subcordiformis* biomass growth unit was comparable, reaching $14.26 \pm 1.20$ $cm^3/g_{ODM}$

in V1 and $11.49 \pm 0.51$ cm$^3$/g$_{ODM}$ in V2 (Figure 4). The technological performance obtained in ST2SE2 was comparable ($p = 0.05$) (Table 4). In V1, the biomass produced $106.32 \pm 10.54$ cm$^3$ of the biogas with the rate of $r = 0.89 \pm 0.09$ cm$^3$/h, whereas the respective values recorded in V2 were at $135.71 \pm 2.17$ cm$^3$ and $r = 1.13 \pm 0.02$ cm$^3$/h (Figures 3 and 5).

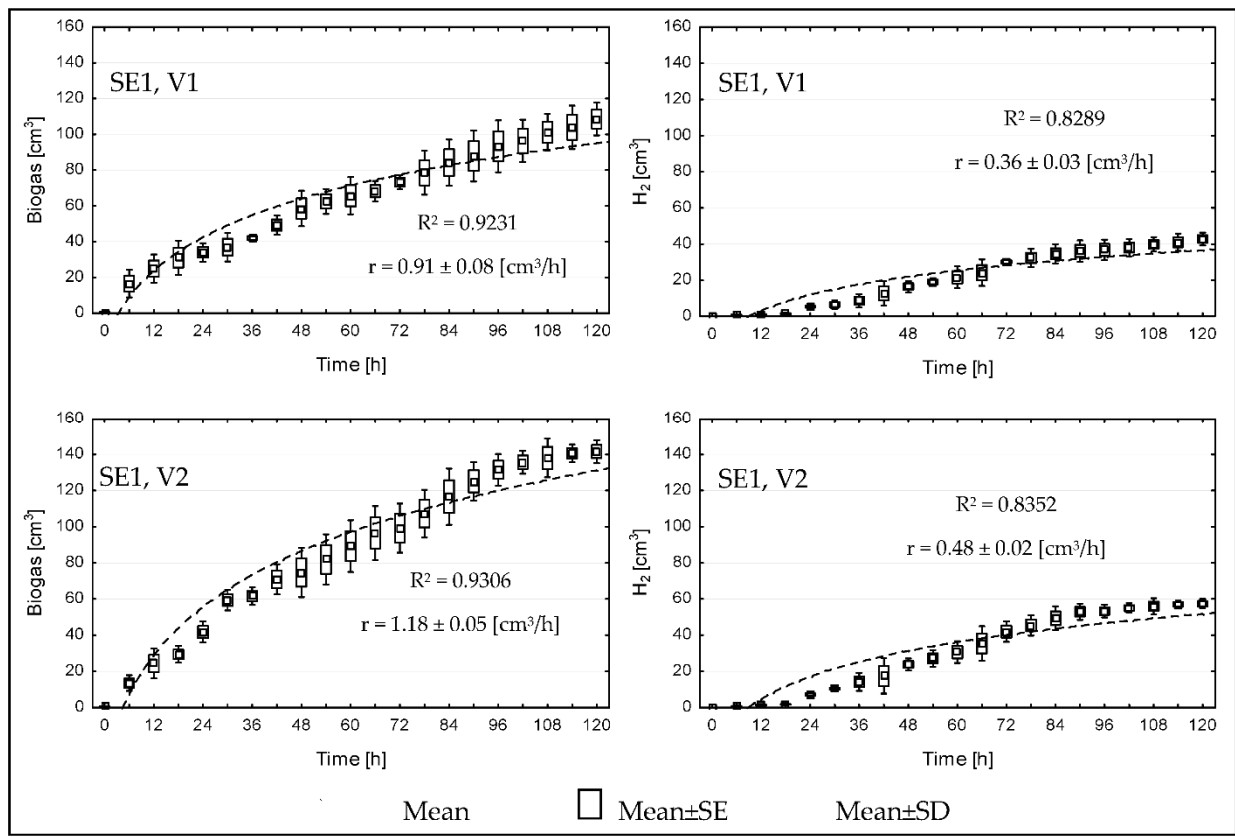

**Figure 2.** The course of biogas and hydrogen production by *P. subcordiformis* biomass in series 1.

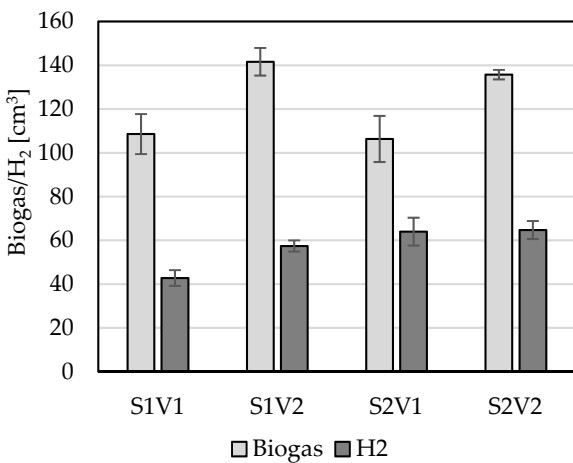

**Figure 3.** Volumes of biogas and hydrogen produced in subsequent experimental series and variants of STAGE 2.

**Table 3.** Qualitative composition of biogas.

| Series | Variant | H$_2$ [%] | CO$_2$ [%] | O$_2$ [%] |
|---|---|---|---|---|
| SE1 | V1 | 41.0 ± 1.4 | 55.3 ± 1.8 | 3.7 ± 0.2 |
|     | V2 | 40.2 ± 1.4 | 57.3 ± 1.7 | 2.5 ± 0.1 |
| SE2 | V1 | 59.9 ± 1.6 | 36.5 ± 1.3 | 3.6 ± 0.2 |
|     | V2 | 63.2 ± 1.4 | 34.6 ± 1.2 | 2.2 ± 0.1 |

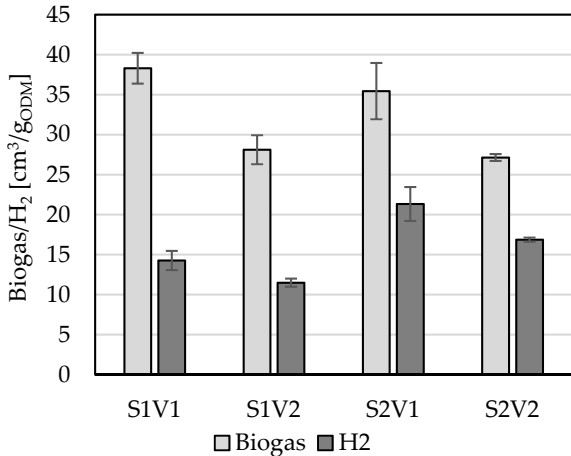

**Figure 4.** Volumes of biogas and hydrogen produced in subsequent experimental series and variants of STAGE 2 per biomass unit.

**Table 4.** The results of a statistical comparative analysis of variables tested in ST2, conducted with HSD Tukey test.

| Variant | Total concentration of biogas produced | | | | Variant | Total concentration of biogas produced per g$_{ODM}$ | | | |
|---|---|---|---|---|---|---|---|---|---|
|  | SE1V1 | SE1V2 | SE2V1 | SE2V2 |  | SE1V1 | SE1V2 | SE2V1 | SE2V2 |
| **SE1V1** |  | *0.013492* | 1.000000 | 0.070077 | SE1V1 |  | 0.050933 | 1.000000 | *0.015422* |
| **SE1V2** | *0.013492* |  | *0.006948* | 0.999733 | SE1V2 | 0.050933 |  | 0.103722 | 0.999989 |
| **SE2V1** | 1.000000 | *0.006948* |  | *0.037985* | SE2V1 | 1.000000 | 0.103722 |  | *0.033422* |
| **SE2V2** | 0.070077 | 0.999733 | *0.037985* |  | SE2V2 | *0.015422* | 0.999989 | *0.033422* |  |
| **Variant** | Total concentration of hydrogen produced | | | | **Variant** | Total concentration of hydrogen produced per g$_{ODM}$ | | | |
|  | SE1V1 | SE1V2 | SE2V1 | SE2V2 |  | SE1V1 | SE1V2 | SE2V1 | SE2V2 |
| **SE1V1** |  | 0.611181 | *0.000816* | 0.690782 | SE1V1 |  | 0.153253 | *0.008029* | *0.000143* |
| **SE1V2** | 0.611181 |  | *0.000145* | *0.015779* | SE1V2 | 0.153253 |  | 0.960106 | *0.000548* |
| **SE2V1** | *0.000816* | *0.000145* |  | 0.074735 | SE2V1 | *0.008029* | 0.960106 |  | *0.011987* |
| **SE2V2** | 0.690782 | *0.015779* | 0.074735 |  | SE2V2 | *0.000143* | *0.000548* | *0.011987* |  |

| Variant | % content of hydrogen in biogas | | | |
|---|---|---|---|---|
|  | SE1V1 | SE1V2 | SE2V1 | SE2V2 |
| **SE1V1** |  | 0.999974 | *0.000143* | *0.000143* |
| **SE1V2** | 0.999974 |  | *0.000143* | *0.000143* |
| **SE2V1** | *0.000143* | *0.000143* |  | 0.523867 |
| **SE2V2** | *0.000143* | *0.000143* | 0.523867 |  |

*Values in italics denote differences significant at p ≤ 0.05*

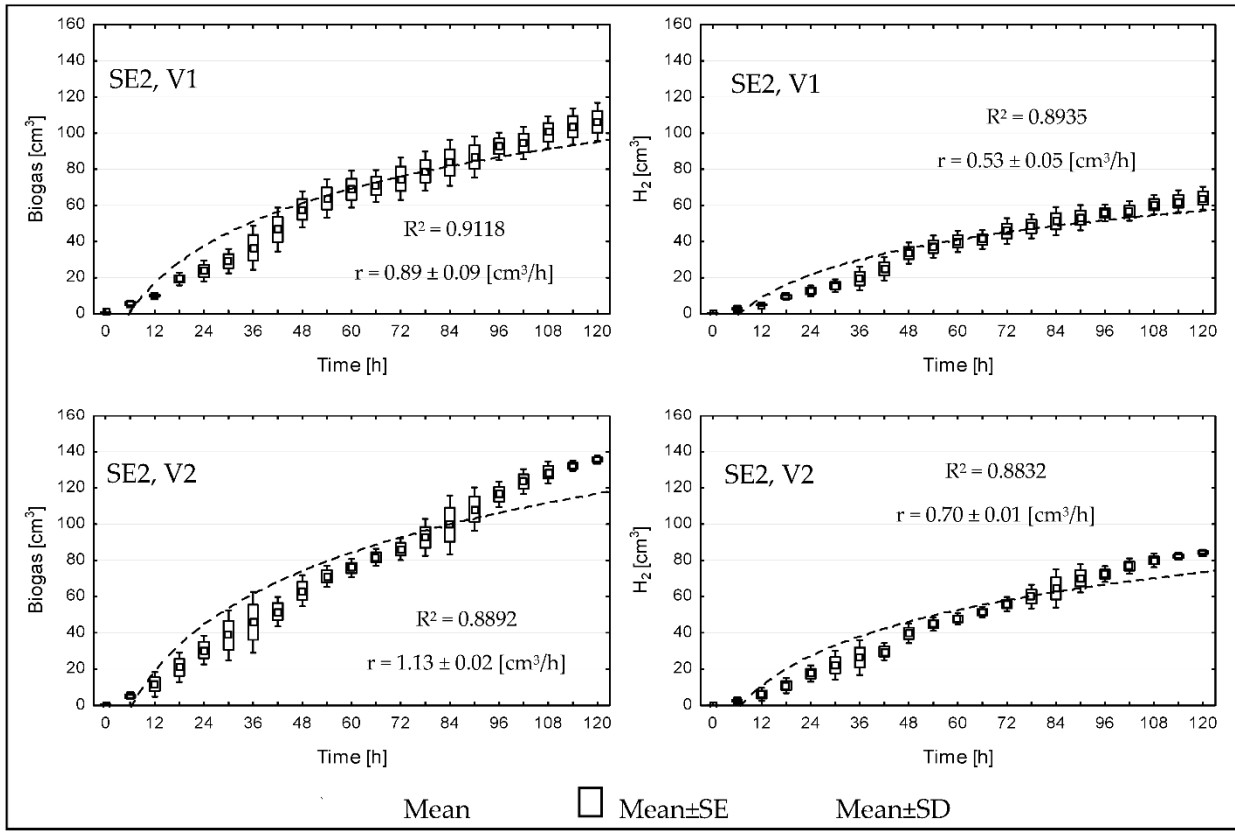

**Figure 5.** The course of biogas and hydrogen production by *P. subcordiformis* biomass in series 2.

The volumes of hydrogen produced were similar in both variants and reached $63.98 \pm 6.35$ cm$^3$ at mean r = $0.53 \pm 0.05$ cm$^3$/h in V1 as well as $64.74 \pm 4.11$ cm$^3$ at r = $0.70 \pm 0.01$ cm$^3$/h in V2 (Figures 3 and 4). The percentage concentration of hydrogen in the biogas differed significantly ($p = 0.05$) between variants (Table 4) and reached $59.9 \pm 1.6\%$ in V1 and $63.2 \pm 1.4\%$ in V2 (Table 3). The technological parameters tested caused statistically significant ($p = 0.05$) differences in the effectiveness of gaseous metabolite production by microalgal biomass per biomass growth unit (Table 4), which reached $35.44 \pm 3.52$ cm$^3$/g$_{ODM}$ in V1 and $27.14 \pm 0.43$ cm$^3$/g$_{ODM}$ in V2 (Figure 5). Biohydrogen production effectiveness was analogous, amounting to $21.33 \pm 2.12$ cm$^3$/g$_{ODM}$ and $16.87 \pm 0.27$ cm$^3$/g$_{ODM}$ in V1 and V2, respectively (Figure 5).

## 4. Discussion

Hydrogen production in bio-processes run by microalgae is based on the direct bio-photolysis, namely a reaction catalyzed by hydrogenase [26]. It proceeds in transmembrane peptidic complexes, the so-called photosystems (PS) [27]. The exposure of these structures to solar radiation results in the water molecule breakdown. One of the PS produces $O_2$, while the other uses the resulting electrons to reduce $CO_2$ and build biomass (aerobic conditions). Electrons can be transferred to hydrogenase via ferredoxin and take part in the production of hydrogen (anaerobic conditions). The anaerobic conditions have been proven to be the most favorable for hydrogen production, with $O_2$ concentration in the medium not exceeding 0.1% [28]. In the presented study, the culture medium in ST2 was deoxidized by efficient nitrogen purging.

For $H_2$ production conditions, the protocol of sulfur deprivation has been applied. By reducing photosynthetic activity, this protocol enables the problem of the high sensitivity of the Fe-hydrogenase to $O_2$ to be get round. The transition to anoxic conditions is then realized as the $O_2$ consumption by respiration process becomes higher than the $O_2$ released during photosynthesis. $H_2$ is then produced under light conditions. This biochemical

mechanism of this process explains the low oxygen concentration in the gas produced [29]. Only few studies present the percentage composition of the gas mixture generated during hydrogen production by microalgae. The results obtained by the authors also confirm the investigations of $H_2$ production using the green microalga Chlamydomonas reinhardtii in a fully controlled photobioreactor fitted with on-line gas analysis [30]. The production of bio-hydrogen by this species takes place under conditions analogous to those used for *Platymonas subcrodiformis*. The studies showed that the progressive decline in $O_2$ production was directly correlated with the decline in PSII activity. This is a major feature and a well-known outcome of the sulfur deprivation protocol [31]. Since mitochondrial respiration is almost constant [32], a decrease in photosynthetic activity disrupts the balance between the two processes, leading to a progressive decrease in $O_2$ concentration. Ultimately, anaerobic conditions are achieved that enable the synthesis of Fe-hydrogenase and the release of $H_2$.

In the research carried out so far, the release of $CO_2$ has also been observed, which is explained by the biodegradation of the reserves of carbon compounds accumulated in microalgae cells. It is accompanied by the appearance of formate in the medium, which is a product of fermentation metabolism [33]. It has been proven that one spare material remaining in the cells is starch, and easily digestible acetate is used [30]. A change in the physiological state of cells is observed, as evidenced by the degradation of pigments, proteins and total sugars [34]. A significant reduction in the value of TOC (total organic carbon) in biomass has also been proven, a significant part of which was metabolized and converted to $CO_2$ [30].

The research presented above has demonstrated that PS is responsible for hydrogen production in conditions where the medium lacks sulfur compounds. For this reason, in the second stage of the present research, sulfur was replaced with chlorine compounds in the culture medium. Usually, the culture medium is deprived of sulfur compounds by using the algae culture centrifugation process, and then suspending the concentrated and liquid phase-free biomass in the medium in which sulfur has been replaced with chlorine compounds [35,36]. It has been proven that the technological treatment based on centrifugation is expensive, time-consuming and leads to partial destruction of the cellular material. An alternative solution in this case is to dilute the culture medium, which directly reduces the sulfur concentration in the technological system. However, it is a method that extends the time needed for sulfur exhaustion and creation of anaerobic conditions [37]. Therefore, the membrane method was deployed in the presented research to separate the microalgal biomass from the culture medium. This alternative method for *P. subcordiformis* biomass dehydration was expected to be less expensive and to ensure higher technological effects.

Some problems may also be encountered with defining cultivation duration and selecting the moment of initiating the phase in which the hydrogen production will take place. Some authors state that the biomass production process should be carried out to the half of the exponential growth phase [38,39]. Others, in turn, argue that the higher density of algal cells directly improves the efficiency and extends the time of hydrogen production [40,41]. Jᵢ et al. proved that with a cell density of 0.5 g/dm$^3$, they achieved hydrogen production at 16 cm$^3$/g biomass, while increasing cell concentration to 3.2 g/dm$^3$ ultimately led to the production of over 49 cm$^3$ $H_2$/g biomass. Along with the density of the substrate, the rate of gas production also increased almost 10 times [42]. Our study did not confirm this phenomenon, regardless of the *P. subcordiformis* biomass concentration applied.

*Chlamydomonas reinhardtii*, which is common in soil and saline waters [43], is a species frequently used for hydrogen production. Studies have reported that its $H_2$ yield reaches 180 cm$^3$/dm$^3$ of the active volume of the bioreactor [15,44]. Faraloni et al. (2011) achieved hydrogen production at 150 cm$^3$/dm$^3$ of the *Chlamydomonas reinhardtii* algae culture using wastewater from the processing of olives in the algae growth process [45]. In turn, Skjanes et al. (2008) investigated the possibility of producing hydrogen from 21 species of green algae. The highest hydrogen yield, approximating 140 cm$^3$/dm$^3$, was reported for *C. reinhardtii*, followed by 80 cm$^3$/dm$^3$ for *C. noctigama* and 22 cm$^3$/dm$^3$ for *C. eu-*

*ryale* [46]. The algae of the genus *Chlorella* sp. have also been shown to be highly potent hydrogen producers [47]. The advantages of this species are due to its eurybiontic character, high adaptability to changing environmental conditions, resistance to pollution and a fast growth rate [48]. Zhang et al. (2014) investigated the effect of depleting the medium in nutrients on the efficiency of hydrogen production by the *Chlorella protothecoides* species algae. The results showed that with nitrogen deficiency, gas production was achieved at 110.8 $cm^3/dm^3$ of culture. Due to the limitation of the concentration of two culture medium components, namely nitrogen and sulfur, the hydrogen production efficiency increased to the value of 140.4 $cm^3/dm^3$ of the culture [49]. In the study by Song et al. (2011), hydrogen production by *Chlorella* sp. ranged from 260 to 480 $cm^3/dm^3$, and the highest technological effects were achieved at temperatures of 37–40 °C [50]. Other publications report on the efficiency of biohydrogen production by *P. subcordiformis* [51,52]. Ji et al. (2011) investigated *P. subcordiformis* productivity in nitrogen, sulfur and phosphorus depleted media. With the same cell density used in different variants of the experiment ($6 \times 10^6$ cells/$cm^3$), the hydrogen production efficiency peaked when *P. subcordiformis* cells were kept in the nitrogen-free medium for 6 days, reaching 55.8 $cm^3$ $H_2/dm^3$ of culture under carbonyl cyanide m-chlorophenylhydrazone (CCCP) protocol [51]. In turn, Guo et al. (2016) obtained the $H_2$ production of $78 \pm 5$ $cm^3/dm^3$ in a photobioreactor with an integrated alkaline fuel cell (AFC), after 40 h of continuous irradiation, which was 1.5 times higher than the value achieved in the algae culture without integrated AFC ($50 \pm 3$ $cm^3/dm^3$) [52].

## 5. Conclusions

Given the weaknesses of conventional methods for hydrogen production, biological methods, including the process of direct biophotolysis taking place in microalgae cells, are becoming a viable alternative in this respect. The broad range of suitable species and process conditions speaks in favor of this technology. In the presented experiments, variable environmental conditions were deployed to achieve a high algal biomass concentration and hydrogen yield.

The research proved that the efficiency of *P. subcordiformis* biomass production was similar regardless of the culture medium applied. Biomass grown in water from the Bay of Gdańsk was characterized by a significantly higher concentration of hydrogen in the biogas and the total efficiency of hydrogen production. The higher concentration of biomass in the reactors was found to directly increase the total volumes of biogas and hydrogen produced. However, there was no significant influence of the initial *P. subcordiformis* biomass concentration on the hydrogen yield per biomass unit. Apart from hydrogen, the biogas contained carbon dioxide and small amounts of oxygen.

**Author Contributions:** Conceptualization, M.D. (Marcin Dębowski) and M.D. (Magda Dudek); methodology, M.D. (Marcin Dębowski) and M.D. (Magda Dudek); software, M.Z. and A.N.; validation, M.D. (Marcin Dębowski), M.D. (Magda Dudek) and J.K.; formal analysis, M.Z.; investigation, M.D. (Marcin Dębowski); resources, M.D. (Marcin Dębowski) and J.K.; data curation, M.D. (Magda Dudek) and A.N.; writing—original draft preparation, M.D. (Marcin Dębowski), M.D. (Magda Dudek) and J.K.; writing—review and editing, M.D. (Marcin Dębowski), M.D. (Magda Dudek), A.N., M.Z. and J.K.; visualization, M.D. (Marcin Dębowski) and J.K.; supervision, A.N. and M.Z.; project administration, M.Z.; funding acquisition, M.D. (Marcin Dębowski), M.Z. and J.K. All authors have read and agreed to the published version of the manuscript.

**Funding:** The manuscript was supported by Project financially supported by Minister of Education and Science in the range of the program entitled "Regional Initiative of Excellence" for the years 2019–2022, project No. 010/RID/2018/19, amount funding 12.000.000 PLN.

**Institutional Review Board Statement:** Not applicable.

**Informed Consent Statement:** Not applicable.

**Data Availability Statement:** Not applicable.

**Conflicts of Interest:** The authors declare no conflict of interest.

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
