# Peer review of "The Effect of Autotrophic Cultivation of Platymonas subcordiformis in Waters from the Natural Aquatic Reservoir on Hydrogen Yield"

_resources, doi:10.3390/resources11030031_

Round 1
Reviewer 1 Report
I believe that the topic of the manuscript is interesting and beneficial for the readers of the journal. However, the following issues need to be addressed to avoid misunderstanding and improve the messages being delivered:
- In line 91, Table 1 does not provide information about the final concentration of the biomass for ST2.
- In line 101, the authors mention sampling the Bay water for roughly 6 months. However, the following paragraph does not explain which sample (or water taken from a particular time) was used to conduct the experiment. Also, it would be great to explain what were the variation in the water quality during this period in terms of the major operating parameters such as salt concentrations and/or pH.
- A more detailed description on how the chlorophyl, P, and N content were measured is necessary, especially given there are no references on the analytical methods used (section 2.4).
- In Fig. 3, the authors tried fitting the data points with a nonlinear line. There should be explanation on why a particular type of line was used to fit the data instead of, for instance, a line.
- It is difficult to believe that H2 concentration went as high as 42% in the tanks. This is because such a high H2 concentration (even lower than that) will drive the hydrogenases inside the cells to oxidize the gas into protons and electrons. Also, the gas composition in Table 3 is not convincing. There is no N2 in the gas, and the CO2 level is so high which can make the media highly acidic. The O2 concentration is also too low compared to the H2 given how the water splitting will result in one molecule of O2 per 2 moles of H2.
Author Response
Detailed response to remarks of Reviewer #1:
Reviewer’s comment 1:
I believe that the topic of the manuscript is interesting and beneficial for the readers of the journal. However, the following issues need to be addressed to avoid misunderstanding and improve the messages being delivered.
Response:
Authors are grateful for all remarks and suggestions made by Reviewer regarding the manuscript entitled: “The effect of autotrophic cultivation of Platymonas subcordiformis in waters from the natural aquatic reservoir on hydrogen yield” (resources-1617478). All of them were considered in the revised manuscript. In addition, the manuscript was corrected in terms of editorial standards as well as brevity of English language. Below find, please, a detailed response to Reviewer comments.
With kind regards, Authors
Reviewer’s comment 2:
In line 91, Table 1 does not provide information about the final concentration of the biomass for ST2.
Response:
The authors are grateful for this remark. In table 1 is presented the organization of the experiment and the initial (not final) concentration of Platymonas subcordiformis microalgae biomass depending on the variant (V) in stage 2 (ST2). The production of the microalgae biomass was analyzed in stage 1 (ST1) and the final concentration of biomass is presented in Table 2. In stage 2 (ST2) experiments was focused was only on the efficiency of hydrogen production and changes in biomass concentration in the culture medium were not analyzed. Under stressful conditions, the biomass does not grow (possible reduction of population), but the metabolism of microalgae is changed.
Reviewer’s comment 3:
In line 101, the authors mention sampling the Bay water for roughly 6 months. However, the following paragraph does not explain which sample (or water taken from a particular time) was used to conduct the experiment. Also, it would be great to explain what were the variation in the water quality during this period in terms of the major operating parameters such as salt concentrations and/or pH.
Response:
Authors thank a lot the Reviewer for this comment. The research was carried out in four replications (this information was supplemented in the Methodology section, in chapter 2.4. Analytical and statistical methods). Water obtained from the Gdansk Bay in May, July, August and October was used in the experiments. The physico-chemical characteristics of the water in this period was stable and value of the analyzed parameters were very similar (as indicated by the average values with standard deviations presented in lines 97 – 100). Additionally, the water was filtered through filters for qualitative analyses (medium size, Ø 12.5, Eurochem), and then sterilized in a Tuttnauer 2840 EL - D autoclave at a temperature of 121° C for 15 min. This allowed for the removal of other planktonic organisms, possible toxins and the water disinfection was provided.
Reviewer’s comment 4:
A more detailed description on how the chlorophyl, P, and N content were measured is necessary, especially given there are no references on the analytical methods used (section 2.4).
Response:
The authors are grateful for pointing out the lack of important information in methodology section. Additions suggested by the Reviewer were introduced: „ Chlorophyll was determined spectrophotometrically after extraction with 90% acetone. Total nitrogen (EN ISO 11905-1), ammonia nitrogen (ISO 7150-1), total phosphorus (ISO 6878_2004), orthophosphates (ISO 6878_2004), sulphates (ISO 10304-1), chlorine compounds (ISO 9297:1994), iron compounds (DIN 38406-E1), and COD (ISO 6060-1989) were determined using Hach Lange cuvette tests and an UV/VIS DR 5000 spectrophotometer. The same methodology was applied to monitor the levels of essential nutrients in the culture, i.e. Ntot. and Ptot”.
Reviewer’s comment 5:
In Fig. 3, the authors tried fitting the data points with a nonlinear line. There should be explanation on why a particular type of line was used to fit the data instead of, for instance, a line.
Response:
Nonlinear regression is used to determine biogas curves. Iteration method was applied, in which in every iterative step the function is replaced by linear differential in relation to the defined parameters. The φ2 contingency coefficient was adopted as a measure of curve fit (with defined parameters) into test data. This coefficient is a ratio of sum of the squared deviations of the values calculated on the basis of determined function from experimental values, to the sum of squared deviations of experimental values from the mean value. Contingency is the better, the lower is φ2 coefficient. Such model fit to experimental points was assumed in which contingency coefficient did not exceed 0.2.
Reviewer’s comment 6:
It is difficult to believe that H2 concentration went as high as 42% in the tanks. This is because such a high H2 concentration (even lower than that) will drive the hydrogenases inside the cells to oxidize the gas into protons and electrons. Also, the gas composition in Table 3 is not convincing. There is no N2 in the gas, and the CO2 level is so high which can make the media highly acidic. The O2 concentration is also too low compared to the H2 given how the water splitting will result in one molecule of O2 per 2 moles of H2.
Response:
The authors thank the Reviewer for this critical remark and doubt. The gas was analyzed at the end of respirometric measurements using a gastight syringe (20 mL injection volume) and a gas chromatograph (GC, 7890A Agilent) equipped with a thermal conductivity detector (TCD) - in which the measurement is carried out by analyzing changes in electrical conductivity resulting from changes in the thermal conductivity of the atmosphere around the thermocouple when the tested chemical compounds appear in the carrier gas (helium). The magnitude of changes in thermal conductivity was directly proportional to the concentration of the tested gas components. The GC was fitted with the two Hayesep Q columns (80/100 mesh), two molecular sieve columns (60/80 mesh), and Porapak Q column (80/100) operating at a temperature of 70° C. The temperature of the injection and detector ports were 150° C and 250° C, respectively. Helium and argon were used as the carrier gases at a flow of 15 mL/min.
The authors emphasize that the percentage values ​​only illustrate the qualitative composition of the gases mixture. The absolute amounts were low and are shown in Figures 2-5. The observed absolute concentrations did not adversely affect the metabolism of microalgae. In the composition of gases presented in Table 3, nitrogen (N2) was not included because it was artificially introduced into the bioreactors in order to remove oxygen from the culture medium. The percentages of the remaining gases have been converted proportionally according to the equation, e.g. for hydrogen::
H2[%] = H2[%] (in gas with N2) x (1 - N2[%]/100%).
In order to explain to the reader the phenomena occurring in the cultivation and production of hydrogen by microalgae, the following passage has been added to the discussion:
„For H2 production conditions, the protocol of sulfur deprivation has been applied. By reducing photosynthetic activity, this protocol enables the problem of the high sensitivity of the Fe-hydrogenase to O2 to be get round. The transition to anoxic conditions is then realized as the O2 consumption by respiration process becomes higher than the O2 released during photosynthesis. H2 is then produced under light conditions. This biochemical mechanism of this process explains the low oxygen concentration in the gas produced [1]. Only few studies present the percentage composition of the gas mixture generated during hydrogen production by microalgae. The results obtained by the authors also confirm the investigations of H2 production using the green microalga Chlamydomonas reinhardtii in a fully controlled photobioreactor fitted with on-line gas analysis [2]. The production of bio-hydrogen by this species takes place under conditions analogous to those used for Platymonas subcrodiformis. The studies showed that the progressive decline in O2 production was directly correlated with the decline in PSII activity. This is a major feature and a well-known outcome of the sulfur deprivation protocol [3]. Since mitochondrial respiration is almost constant [4], a decrease in photosynthetic activity disrupts the balance between the two processes, leading to a progressive decrease in O2 concentration. Ultimately, anaerobic conditions are achieved that enable the synthesis of Fe-hydrogenase and the release of H2.
In the research carried out so far, the release of CO2 has also been observed, which is explained by the biodegradation of the reserves of carbon compounds accumulated in microalgae cells. It is accompanied by the appearance of formate in the medium, which is a product of fermentation metabolism [5]. It has been proven that one spare material remaining in the cells is starch, and easily digestible acetate is used [2]. A change in the physiological state of cells is observed, as evidenced by the degradation of pigments, proteins and total sugars [6]. A significant reduction in the value of TOC (total organic carbon) in biomass has also been proven, a significant part of which was metabolized and converted to CO2 [2]”.
- Ghirardi, M.L.; Zhang, L.; Lee, J.W.; Flynn, T.; Seibert, M.; Greenbaum, E.; Melis, A. Microalgae: a green source of renewable H(2). Trends Biotechnol. 2000, 18(12), 506-11. https://doi.org/10.1016/s0167-7799(00)01511-0.
- Fouchard, S.; Pruvost, J.; Degrenne, B.; Legrand, J. Investigation of H2 production using the green microalga Chlamydomonas reinhardtii in a fully controlled photobioreactor fitted with on-line gas analysis. Int J Hydrog Energy 2008, 33, 3302–3310. https://doi.org/10.1016/j.ijhydene.2008.03.067.
- Wykoff, D.D.; Davies, J.; Melis, A.; Grossman, A.R. The regulation of photosynthetic electron transport during nutrient deprivation in Chlamydomonas reinhardtii. Plant Physiol. 1998, 117, 129–139. https://doi.org/10.1104/pp.117.1.129.
- Melis, A.; Zhang, L.; Forestier, M.; Ghirardi, M.L.; Seibert, M. Sustained photobiological hydrogen gas production upon reversible inactivation of oxygen evolution in the green alga Chlamydomonas reinhardtii. Plant Phys. 2000, 122, 127–136. https://doi.org/10.1104/pp.122.1.127.
- Gfeller, R.P.; Gibbs, M. Fermentative metabolism of Chlamydomonas reinhardtii: I. Analysis of fermentative products from starch in dark and light. Plant Physiol. 1984, 75, 212–218. https://doi.org/10.1104/pp.75.1.212.
- Ghirardi, M.L.; Kosourov, S.; Tsygankov, A.; Rubin, A.; Seibert, M. Cyclic photobiological algal H2-production. In: Proceedings of the 2002 U.S. DOE Hydrogen Program Review. NREL/CP-610-32405 2002, 1–12.
Reviewer 2 Report
Please see the attachment

Author Response
Detailed response to remarks of Reviewer 2:
Reviewer comment 1:
The subject discussed in this paper is an important and interest to the readers of the Resources journal. I recommend that the manuscript be approved for publication in journal, after the Minor revisions. Some of the corrections identified are given below.
Response:
The authors are very grateful to the Reviewer for a deep verification of the manuscript entitled: “The effect of autotrophic cultivation of Platymonas subcordiformis in waters from the natural aquatic reservoir on hydrogen yield” (ID resources-1617478), appropriate and constructive suggestions to improve our paper. We have addressed all the issues raised and have modified our manuscript accordingly. We hope that the changes we performed and our responses will be satisfied to the Reviewer. With kind regards, Authors
Reviewer comment 2:
Page 4: Line 106: "[...]" - what does this sign mean?
Response:
The authors thank the Reviewer for pointing out this technical mistake. It has been corrected.
Reviewer comment 3:
Page 4: please use the identical name of the reactor: "BioFlo115" or with the space "BioFlo 115"
Response:
Corrected as suggested by the Reviewer.
Reviewer comment 4:
Please explain the abbreviations the first time you use them, e.g. COC, ODM etc. For example, “organic dry matter (odm)” - page 4, line 131. Please keep the letter size for the abbreviation "3.0 gODM/dm3 "; and page 7, line 191 ” 0.43 cm3 /gODm”
Response:
The authors thank the Reviewer. Corrected agree with Reviewer suggestion.
Reviewer’s comment 5:
In Figures 2 and 3, the decimal separator needs to be corrected. There is "," instead of "." Response:
The authors thank the Reviewer for pointing out this technical mistake. It has been corrected in line with suggestion.
Reviewer’s comment 6:
(Figure 2,4)" or "(Figure 3,4)" are difficult to identify. May use a different numbering, e.g. 2 (d), 3 (d)
Response:
Thank you for the reviewer's suggestion. The authors believe that the markings used in the charts are legible. The used axle descriptions as well as the series and variants designations enable an accurate reading.
Reviewer’s comment 7:
In the last sentence of the conclusions, the authors state that the produced biogas contains trace amounts of oxygen. About three percent of the oxygen in biogas is, in my opinion, a small amount, and I would not recommend the word "trace", often expressed in smaller units such as ppm, for CO or H2S
Response:
We agree with the Reviewer's remark. This has been corrected.